# More than a Physical Problem: The Effects of Physical and Sensory Impairments on the Emotional Development of Adults with Intellectual Disabilities

**DOI:** 10.3390/ijerph192417080

**Published:** 2022-12-19

**Authors:** Paula S. Sterkenburg, Marie Ilic, Miriam Flachsmeyer, Tanja Sappok

**Affiliations:** 1Department of Clinical Child and Family Studies & Amsterdam Public Health, Faculty of Behavioural and Movement Sciences, Vrije Universiteit Amsterdam, Van der Boechorststraat 7, 1081 BT Amsterdam, The Netherlands; 2Department of Assessment and Treatment, Bartiméus, 3941 XM Doorn, The Netherlands; 3Diakonische Stiftung Wittekindshof, 32549 Bad Oeynhausen, Germany; 4Berlin Center for Mental Health in Intellectual Developmental Disabilities, Ev. Krankenhaus Königin Elisabeth Herzberge, 10365 Berlin, Germany

**Keywords:** emotional development, intellectual disability, visual impairment, hearing impairment, physical disability, sensory impairments

## Abstract

With the introduction of the ICD-11 and DSM-5, indicators of adaptive behavior, including social–emotional skills, are in focus for a more comprehensive understanding of neurodevelopmental disorders. Emotional skills can be assessed with the Scale of Emotional Development-Short (SED-S). To date, little is known about the effects of physical disorders and sensory impairments on a person’s developmental trajectory. The SED-S was applied in 724 adults with intellectual disabilities, of whom 246 persons had an additional physical and/or sensory impairment. Ordinal regression analyses revealed an association of movement disorders with more severe intellectual disability and lower levels of emotional development (ED) on the overall and domain levels (*Others*, *Body*, *Material*, and *Communication*). Visual impairments predicted lower levels of ED in the SED-S domains *Material* and *Body*, but not the overall level of ED. Hearing impairments were not associated with intellectual disability or ED. Epilepsy correlated only with the severity of intellectual disability. Multiple impairments predicted more severe intellectual disabilities and lower levels of overall ED. In conclusion, physical and sensory impairments may not only affect physical development but may also compromise intellectual and emotional development, which should be addressed in early interventions.

## 1. Introduction

People with intellectual disabilities have a higher risk of developing psychological stress than people without intellectual disabilities [1]. Possibly contributing to this, people with intellectual disabilities have difficulty assessing and processing information [2], they need predictable environments [3], and they have poor coping mechanisms [4]. Furthermore, due to affected children’s limited behavioral repertoires, parents and caregivers need to have a high level of sensitivity and responsiveness to signals and react adequately to the behavior, needs, and wishes of the person with an intellectual disability [4,5]. Consequently, there is a higher risk of disturbed attachment and emotional and behavioral problems [6,7].

According to the American Association on Intellectual and Developmental Disabilities (AAIDD), people are classified as having an intellectual disability when they have an IQ score below 70 and/or a significant limitation in their adaptive behavior [8]. The need for support for conceptual, social, and practical skills starts before the age of 22 [8]. Likewise, the classification systems DSM-5 and ICD-11 also include limitations in adaptive behavior and social skills in the diagnostic criteria for intellectual disability/disorders of intellectual development, respectively. Intellectual disability is often associated with a delay in emotional development [9]. Delayed social–emotional development and associated neglect of basic emotional and support needs result in stress and consequently possibly challenging behavior [7]. To provide adequate care matching the emotional functioning and needs of persons with intellectual disability, the Scale for Emotional Development-Short (SED-S) has been developed and studied in children with and without intellectual disabilities [9,10,11] as well as in adults [12,13,14,15]. The SED-S is based on the Scheme for Appraisal of Emotional Development (SAED) [16], the Scale for Emotional Development–Revised (SED-R) [17], and the SED-R2 [18]. Adequate psychometric properties are reported for the SED-S [9,10].

The prevalence of co-occurring impairments (such as physical, visual, and hearing impairments) is higher among persons with intellectual disability than in those without [19,20]. In studies conducted in the Netherlands among 1598 persons with intellectual disability who were older than 18 years of age, the following prevalence rates were reported: visual impairment, 14%; blindness, 5%; mild hearing loss, 30%; moderate to severe hearing loss, 15%; and dual impairments (visual and hearing impairments), 5% [20,21,22]. What is more, in a Finnish study among 461 persons with profound to severe intellectual disability, the prevalences of motor impairments and epilepsy were, respectively, 35% and 51% [23]. These impairments in addition to intellectual disability may affect emotional functioning.

Auditory sensory loss is, for example, of interest when studying social–emotional development in people with intellectual disabilities. In studies conducted before the implementation of neonatal hearing screening programs, results indicated that children over the age of 4 with hearing loss had more social–emotional difficulties than did children without hearing loss [24]. An uncompensated hearing impairment also affects communication in a negative way in adulthood [25,26]. However, the results of a study conducted in 18-month-old children with hearing loss found no apparent difficulties in social–emotional functioning [27]. Additionally, Lederberg and Mobley [28] found no difference in the quality of attachment of children with and without hearing impairments. This could be thanks to the ability of parents and children to communicate nonverbally during the child’s first years, with children with a hearing impairment being able to adequately imitate facial expressions and hand and body movements, in addition to using gestures and pointing [29]. Moreover, parents put much effort into early communication, such as trying to establish (eye) contact or elicit a response through manifesting exaggerated facial expressions, waving their hands, and moving objects or people into the child’s line of sight [29]. Although parents and children have compensatory communication in place, Dirks et al. [30] did report that children with a hearing impairment are at risk for social–emotional difficulties. Adding to this, Sterkenburg et al. [9] reported that children with an intellectual disability and a hearing-and/or visual impairment (n = 9) showed lower emotional functioning than children with intellectual disabilities without these sensory impairments (n = 108). These results were found on the SED-S overall score and on the domain *Others* (meaning “relating to significant others”). However, due to the small sample size in this study, people with visual or hearing impairments were combined into one group. Consequently, it is unclear whether emotional functioning is lowered by the separate sensory impairments.

Regarding the emotional development of persons with visual disabilities, the physical, cognitive, and social development of a child are affected by visual impairment [31]. Urqueta Alfaro et al. [32] confirmed that, compared with children without visual impairment, the overall development of children with visual disabilities is delayed. To be precise, the visual impairment hinders their social development, as social behavior and communication mostly develop through eye contact and observation [31]. Zooming in on the emotional development of children with visual impairment, Fraiberg [33] reported that blind children show more fear of strangers than sighted peers and that their facial expressions are frequently indistinct. Additionally, their nonverbal expressions are sometimes different and very subtle [34]. The findings of Fraiberg [33], suggested that children with a visual impairment do develop mental representations of their attachment figures, but that these representations start at a later age and/or show different characteristics than in sighted peers. This delay may affect their emotional functioning. Furthermore, infants with visual impairments need more help and encouragement to become aware of their range of motion. This help prevents them from remaining in their natural passive state, which could hinder their physical development, including posture and mobility [35], and affect their exploration and social and emotional development.

Next to the influence of a hearing or visual impairment, a child’s physical development may also impact their emotional development. Persons with an intellectual disability have an increased risk for physical disabilities and epilepsy [19,36]. Vandesande et al. [37] conducted a study among children with severe and profound intellectual disabilities during a stressful situation, where the child’s parents and a stranger were present. Interestingly, the child’s differentiated responses to comfort seemed to be related to their fine motor skills [37]. Therefore, to adequately support persons with intellectual and physical disabilities, it is important to examine the effects of the disabilities on the different domains of emotional development.

According to Došen [38], the social–emotional development of a neurotypical child goes through several stages, and these stages are linked to social–emotional developmental milestones [38,39,40]. For children with intellectual disabilities and sensory, physical, or multiple impairments, reaching these milestones requires more effort than it does for children without these impairments. Up to now, to our knowledge, no studies have been conducted among adults with intellectual disability to examine the effect of visual, hearing, or physical impairments on emotional functioning. Therefore, the aim of this study was to examine whether physical or sensory impairments of adults with intellectual disabilities can predict their level of social–emotional functioning.

## 2. Materials and Methods

### 2.1. Setting and Design

The study was conducted from May 2016 to November 2020 in three hospitals and five home care facility centers in Belgium, the Netherlands, and Germany. Inclusion criteria were age >18 years and a diagnosis of intellectual disability. There were no exclusion criteria within this respective population. A sample of 724 adults with intellectual disabilities was recruited, among which 246 persons (34%) had an additional physical disorder and/or sensory impairment such as a hearing and/or visual impairment, movement disorder, and/or epilepsy.

The participating organizations were Tordale in Torhout (Belgium), Cordaan in Amsterdam (Netherlands), ORO in Helmond (Netherlands), De Twentse Zorgcentra in Losser (Netherlands), Bartiméus in Doorn (Netherlands), the Evangelisches Krankenhaus Königin Elisabeth Herzberge in Berlin (Germany), Klinikum München Oberbayern in München (Germany), and St. Lukas-Klink in Liebenau (Germany). The persons themselves or their legal guardians gave their informed consent for participation in this study.

### 2.2. Participants

The total study sample consisted of 724 participants with intellectual disabilities. Due to missing information about the researched impairments, two persons were omitted from the analyses, leaving a total sample of 722 participants. The average participant age was 37.4 years (18–76 years, *SD* = 13.3) and males were slightly in the majority (56.4%). The degree of intellectual disability ranged from mild (IQ 50–55 to 70: 28.2%), moderate (IQ 35–40 to 50–55: 37.4%), and severe (IQ 20–25 to 35–40: 26.8%), to profound (IQ < 20–25: 7.6%).

The age of the 246 (34%) participants with sensory impairments ranged from 18 to 76 years, and the average age was 39.9 years (*SD* = 13.7). This sample included more males (60.2%) than females. All levels of intellectual disability were represented; most of the participants had moderate intellectual disabilities (38.2%), followed by severe intellectual disabilities (28.5%), mild intellectual disabilities (19.5%), and profound intellectual disabilities (13.8%). In this group with sensory impairments, 44 persons (17.9%) had hearing impairments and 64 persons (26%) had visual impairments. In the sample with sensory impairments, 87 persons (35.4%) had a movement disorder. Epilepsy was reported in 130 persons (52.8%).

### 2.3. Assessment

The level of emotional development (ED) was assessed using the SED-S [10]. The SED-S is a semi-structured interview consisting of 200 binary items in eight domains, concerning different aspects of daily life behaviors. The scale assesses five developmental stages with reference ages ranging from 0 to 12 years of age. Each level of ED is assessed by five items per domain. The eight domains are: (1) *Body* (Relating to His/Her Own Body), (2) *Others* (Relating to Significant Others), (3) *Object* (Dealing with *Change*: Object Permanence), (4) *Emotions* (Differentiating Emotions), (5) *Peers* (Relating to Peers), (6) *Material* (Engaging with the Material World), (7) *Communication* (Communicating with Others), and (8) *Affect* (Regulating Affect).

Within the SED-S items, behaviors for a certain level of ED are described, which are either typical in the respective person (“yes” answers) or not typical (“no” answers). To determine the level of ED in a certain domain, the number of “yes “answers is counted. The level with the most “yes “answers is the domain-wise level of ED. Estimating the overall level of ED, these domain-specific results are ordered from low to high, and the four lowest domains determine the overall result. Trained psychologists, psychiatrists, developmental psychologists, or ortho-pedagogues conducted the interview with two to five informants, such as family members or close caregivers. The assessment relied on behaviors displayed during the previous 2 weeks.

Expert validity can be taken as given, as the scale is based on a survey of developmental psychology experts and their assessments of behaviors typical for specific levels of development, [10]. Validation against a group of 160 typically developed children showed a high degree of correspondence (81% exact agreement; 0.95 weighted kappa value) [11]. An exploratory factor analysis provided a one-factor model with a good model fit in 724 adults with ID, most of them having additional mental health problems [15,41], in 118 children with ID and mental health problems [9] and in 83 healthy adults with ID [14,41]. Divergent validity was found for chronological age in children with ID [9] and in healthy adults with ID [14,41]. Convergent validity with the Vineland Adaptive Behavior Scale could be seen in the children’s sample (r = 0.642, *p* < 0.001; [9]). Strong negative associations with the severity of ID could be shown in 327 adults with ID and mental health problems (r = −0.654, *p* < 0.001; [11]), in 83 adults with ID without mental health problems (−0.753, *p* < 0.001; [14,41]) and in 118 children with ID (G = −0.69; *p* < 0.001; [9]). Inter-rater reliability for 25 typically developed children was 1.0 (Cohen’s kappa). Internal consistency as measured by Cronbach’s alpha was 0.99 in typically developing children [11], 0.94 in 118 children with ID [9] and 0.92 in 83 adults with ID without mental health problems [14,41].

The degree of intellectual disability was diagnosed using the Disability Assessment Schedule (DAS), which is an informant-based structured interview [41]. It poses several questions about adaptive behaviors in four parts: continence, self-help skills, communication, and (cognitive) skills. Depending on the presence of these behaviors, points are given and summed to a total score (minimum 15 points; maximum 71). The total score corresponds with the levels of ID (mild, moderate, severe, profound). The DAS is a reliable measure when applied by trained professionals [42]. Evidence pertaining to its validity in people with ID is provided by correlation analysis with the Colored Progressive Matrices and the Columbia Mental Maturity Scale [43].

The data regarding hearing and visual impairment, movement disorders, and epilepsy were systematically recorded upon the assessment of the SED-S.

### 2.4. Statistical Analysis

The statistical analyses were conducted using IBM SPSS 27 Statistics for Windows, USA. Associations of hearing and/or visual impairment, and movement disorder and/or epilepsy, with the severity of the intellectual disability and the level of ED (overall and domain wise) were examined by applying an ordinal logistic regression analysis. The ordinal logistic regression model was chosen because of the dichotomous variable structure of the impairments and because the severity of the intellectual disability and level of ED were ordinal variables [44]. A further key assumption of ordinal logistic regression analysis is that of assumption of proportional odds, which in SPSS is examined with the test of parallel lines. Chi-square was used to determine whether the assumed model with an explanatory variable was improved in comparison with the baseline model without this explanatory variable.

A significant *p*-value indicates an improved fit to the data. Nagelkerke *R*^2^ is reported as a pseudo *R*^2^ value, indicating the proportion of variation in the outcome that can be accounted for by the explanatory variables. To specifically analyze the relationship between the sensory impairments and the intellectual disability and ED, odds ratios and the Wald χ^2^ were calculated to investigate whether a significant impact of the explanatory variables existed.

## 3. Results

Assumptions of parallel lines were met for all our analyses (*p* > 0.05) with one exception: for the regression of the domain *Others* on the four impairments, it was significant with *p* = 0.044, meaning these results should be interpreted cautiously.

### 3.1. Number of Impairments as a Predictor of Intellectual Disability and ED

Since many participants had more than one reported impairment, the number of impairments (e.g., visual, hearing) was analyzed as a predictor of the intellectual disability and ED. For intellectual disability, ordinal regression analysis revealed that higher numbers of co-occurrent physical impairments were related to more severe intellectual disability (Δχ^2^ = 37,85, *df* = 4, *p* < 0.001; Nagelkerke *R*^2^ = 0.055). The individual odds ratios were nonsignificant. Additionally, the level of ED improved the model fit of the regression analysis when the number of impairments was used as a predictor to explain the level of ED (Δχ^2^ = 11.56, *df* = 4, *p* < 0.021; Nagelkerke *R*^2^ = 0.017). Again, the individual odds ratios were nonsignificant.

### 3.2. Different Forms of Impairments as Predictors of Intellectual Disability and ED

Looking at the frequency of the different levels of intellectual disability and ED for the different groups of impairments (reported in Table 1 and Table 2), not only the presence or absence of physical or sensory impairments, but also the type of impairment was relevant. For an overview of the results, see Table 3.

Hearing impairment. Hearing impairments were not associated with the severity of intellectual disability or the level of ED in any of the ten analyses (Details for the nonsignificant analyses are available from the authors upon request).

Visual impairment. Having a visual impairment was related neither to the intellectual disability nor to the level of overall ED. However, visual impairments significantly predicted lower levels of ED on the domains of *Body* (odds ratio 1.65, 95% CI 1.022–2.68; Wald *χ*^2^(1) = 4.082, *p* = 0.043) and *Material* (odds ratio 1.71, 95% CI 1.06–2.72, *Wald χ*^2^(1) = 4.823, *p* = 0.028).

Movement disorder. A movement disorder was related to more severe forms of the intellectual disability (odds ratio 0.28, 95% CI 0.18–0.43; Wald *χ*^2^(1) = 33.446, *p* < 0.001). Furthermore, the presence of a movement disorder was related to lower levels of ED in general (odds ratio 1.59, 95% CI 1.05–2.42; Wald *χ*^2^(1) = 4.815, *p* = 0.028) and for the domains of *Body* (odds ratio 2.25, 95% CI 1.47–3.45; Wald *χ*^2^(1) =13.988, *p* < 0.001), *Others* (odds ratio 1.66, *95% CI* 1.06–2.72; Wald *χ*^2^(1) = 5.591, *p* = 0.018), *Material* (odds ratio 1.66, 95% CI 1.10–2.50; Wald *χ*^2^(1) = 5.821, *p* = 0.016), and *Communication* (odds ratio 2.14, 95% CI 1.41–3.24; Wald *χ*^2^(1) = 12.964, *p* < 0.001).

Epilepsy. Epilepsy was associated with more severe forms of intellectual disability (odds ratio 0.65, 95% CI 0.46–0.93; Wald *χ*^2^(1) = 5.590, *p* = 0.018), but not with the level of ED (neither overall nor domain wise).

## 4. Discussion

The aim of this study was to examine if adults with intellectual disability and physical and/or sensory impairments score lower on the SED-S than adults with intellectual disability but without these additional impairments. At first, the results indicated that an accumulation of impairments predicts lower emotional functioning. This could be explained by one impairment significantly affecting the other. For example, as Fraiberg [35] reported, a visual impairment may influence physical development. This in turn may shape emotional development. What is more, persons with a disability are unable to compensate for the co-occurrent impairment. For instance, Gunther [45] reported that the presence of a visual impairment in addition to an intellectual disability means that vision cannot be used to compensate for the intellectual disability, and vice versa. Therefore, people with visual and intellectual disabilities find it is even more difficult to understand social relationships than do people with only intellectual disabilities [46]. Additionally, as found in this study, in persons with a visual and intellectual disability as well as an added hearing impairment, emotional functioning is significantly lower than it is in adults with intellectual disabilities with either a hearing or a visual impairment (not having both impairments).

A second finding of this study is that hearing impairments seem not to be associated with intellectual disability or emotional development. These findings indicate that although there is a risk for social–emotional difficulties in children [30], the focus on learning to communicate [29] and socially interact may eventually buffer the effect of the hearing impairment on emotional functioning. Early intervention programs focusing on joint engagement and emotional availability can be used for this purpose [47,48].

Third, in this study, no significant effects of visual impairment and intellectual disability on the total score for emotional development were found. However, it is important to note that there were domain-specific outcomes, namely in the domains *Relating to His/Her Own Body* and *Engaging with the Material World*. Zooming in on the items of these domains, the hindering effect of the visual impairment becomes more clear: for example, in the use of tactile senses to explore the world, the need to feel safe in an open space, and requiring more time to explore materials, while contact with others is less evident. This may confirm what Fraiberg [33] suggested, namely that emotional development could be delayed and/or show different characteristics than it does in sighted peers. These outcomes are presumably of great importance for (early) intervention programs, which it seems should be more focused on these aspects of socio-emotional development. Additionally, as for all persons, disharmonious emotional development may affect well-being, and therefore should be noticed [49].

A fourth finding of this study is that having a physical disability was associated with more severe forms of intellectual disability and lower levels of emotional functioning. These results indicate that a tailored approach is needed for these persons. Due to the severe or profound disability, repetition, patience, and interaction are needed for emotional development. Nowadays, technology can also be used in daily care to support emotional development [50] and communication. The results of this study stress the importance of addressing these caregiving and assistive technology needs in interventions.

Fifth, epilepsy was related to intellectual disability but could not be linked to emotional development. Thus, epilepsy does not seem to directly obstruct emotional functioning, although adequate care and support for emotional and physical needs is required.

Sterkenburg et al. [9] found a significant relation between visual and hearing senses and emotional development in children. However, in this study the focus was on adults with a mean age of 37.4 years and less-significant links were found. Matching the findings reported by Fraiberg [33] for children with a visual impairment, there was a delay in development, but there seems to be a catching up later in life. However, accumulated impairments did show significant relations with emotional development. Thus, (early) intervention programs are essential and should consider the different co-occurring impairments. There are programs such as: “Development of an attachment relationship” [51]; “Learning together” [52]; “Little room” [53]; “Barti-mat” [54], and the Light Curtain [55]. These results may encourage the application of early interventional strategies that impact both physical and social–emotional skills in people with intellectual and additional disabilities. Hereby, longer-term outcomes, quality of life, and social participation may be improved substantially. Therefore, we recommend continued investment in and provision of (early) intervention programs for parents of infants with intellectual disability and sensory impairment. Additionally, there is a need for more projects focusing on parent–child interaction and the use of technology in supporting the social and emotional development of persons with intellectual disability and sensory impairment.

### Limitations

During the assessment, the data regarding visual and hearing impairments, physical disabilities, and epilepsy were reported based on participants’ records. However, on-the-spot assessments were not conducted. Thus, the data may be incomplete or not adequately reported in the records and may, therefore, be biased. However, comparing the prevalence of the impairments reported in this study to the prevalence mentioned in the literature [20,21,22,23], this does not seem to be the case.

The reported results are based on cross-sectional data, so causal interpretations need to be made with caution. A longitudinal study that follows the development of children with and without intellectual disabilities and physical impairments would be required to be able to fully understand how these factors influence each other and social–emotional development.

The study was conducted in three Western European countries. Due to the small sample size for each of the studied impairments, comparisons between countries were not possible. There may be differences that we cannot now report and that need to be studied in future research. Furthermore, the (early) interventional care in the studied countries may explain the possible catching up when people have reached adulthood; such catching up may not occur in other parts of the world. Replication of this study is therefore needed, covering a broader spectrum of the world population.

## 5. Conclusions

In summary, higher numbers of co-occurrent physical and/or sensory impairments were related to more severe intellectual disability and lower levels of emotional development (ED). Interestingly, movement disorders affected intellectual and ED, while auditory impairment did not affect ED. Epilepsy correlated only with the degree of intellectual disability but not with the level of ED. On the domain level, *Relating to His/Her Own Body* and *Engaging with the Material World* showed associations with visual impairments and movement disorder, the latter disorder also predicting lower levels of ED in the domains *Relating to Significant Others* and *Communicating with Others.* Considering these overall and specific effects of different impairments on the intellectual and emotional functioning of people with multiple disabilities may align early interventions and thereby further improve long-term outcomes.

## Figures and Tables

**Table 1 ijerph-19-17080-t001:** The Different Impairments and Severities of Intellectual Disability.

Severity of Intellectual Disability	No Additional Impairment (*n* = 476)	Hearing Impairment (*n* = 44)	Visual Impairment (*n* = 64)	Movement Disorder (*n* = 87) *	Epilepsy (*n* = 130) *
Mild (*n* = 204)	153 (32.1)	10 (22.7)	13 (20.3)	11 (12.6)	20 (15.4)
Moderate (*n* = 271)	178 (37.4)	16 (36.4)	24 (37.5)	24 (27.6)	57 (43.8)
Severe (*n* = 194)	124 (26.1)	12 (27.3)	20 (31.3)	30 (34.5)	36 (27.7)
Profound (*n* = 55)	21 (4.4)	6 (13.6)	7 (10.9)	22 (25.3)	17 (13.1)

Note. In parentheses are percentages per column. Regression analyses for the impairments marked with * showed a significant association with the severity of intellectual disability.

**Table 2 ijerph-19-17080-t002:** The Different Impairments and Levels of Emotional Development.

Level of ED	No additional Impairment (*n* = 473)	Hearing Disorder (*n* = 44)	Visual Disorder (*n* = 64)	Movement Disorder * (*n* = 87)	Epilepsy (*n* = 129)
SED-S 1	66 (13.9)	10 (22.7)	14 (21.9)	19 (21.8)	21 (16.2)
SED-S 2	112 (23.3)	6 (13.6)	16 (25)	24 (27.6)	36 (27.7)
SED-S 3	164 (34.5)	18 (40.9)	25 (39.1)	33 (37.9)	52 (40.0)
SED-S 4	114 (23.9)	9 (20.5)	8 (12.5)	10 (11.5)	16 (12.3)
SED-S 5	17 (3.6)	1 (2.3)	1 (1.6)	1 (1.1)	4 (3.1)

Note. In parentheses are percentages per column. Regression analyses for the impairments marked with * showed a significant association with level of ED.

**Table 3 ijerph-19-17080-t003:** The associations between sensory impairments, ID, and ED.

Sensory Impairments	Intellectual Disability (ID)	Emotional Development (ED)
Hearing impairment	No association	No association
Visual impairment	No association	No association -except on domain level for the domains *Body* and *Material*
Movement disorder	Relation to more severe forms of ID	Relation to lower levels of ED -on domain level for *Body*, *Others*, *Material,* and *Communication*
Epilepsy	Relation to more severe forms of ID	No association
Accumulation of impairments	Relation to more severe forms of ID	Related to lower levels of ED

## Data Availability

On request.

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
