# Peer review of "More than a Physical Problem: The Effects of Physical and Sensory Impairments on the Emotional Development of Adults with Intellectual Disabilities"

_ijerph, 2022, doi:10.3390/ijerph192417080_

Round 1

Reviewer 1 Report

Issues commented on

Is it DSM 5 or DSM V?

The introduction section confuses how the impairments are described and their association. It seems like the impairments lead to intellectual disability

sentence no 2 under the introduction instead of saying "experience problems with coping" maybe say "have poor coping mechanisms"

The literature review was focused on children. As much as there is limited literature in this field, build the literature on children to link with how these disabilities then continue to impair the participants even in their adult lives.

The study hypothesis needs to be reviewed to tally with the study aim. Please note it is understood that the Independent variables are sensory and physical impairments and the dependent variable is social-emotional functioning. The intellectual disability mainly specifies the participant population

under participants place the 60.2% just after males to avoid confusing that with the female participants

demonstrate how the SED-S scale is valid and reliable for the adult population as it indicates that it assesses the five developmental stages ranging from 0 to 12 years.

under limitations, it is suggested that instead of saying file studies, rather than write participants' records.

Author Response

Dear reviewer

We appreciate the feedback we received. We are also grateful for the opportunity to revise our manuscript. Below we will respond point-by -point on each reviewer’s comments. In the revised manuscript, revisions are marked in yellow. The language editing was done this is not marked.

  1. Is it DSM 5 or DSM V?

Response: Thank you for this comment. Indeed it should have been DSM 5. We now corrected it.

  1. The introduction section confuses how the impairments are described and their association. It seems like the impairments lead to intellectual disability

Response: To be more clear about the relations between the sensory impairments and ID we added Table 3. See results.

  1. sentence no 2 under the introduction instead of saying "experience problems with coping" maybe say "have poor coping mechanisms"

Response: Thank you for your suggestion. It is now adapted.

  1. The literature review was focused on children. As much as there is limited literature in this field, build the literature on children to link with how these disabilities then continue to impair the participants even in their adult lives.

Response: Indeed, we now added the study conducted by Eisinger, J.; Dall, M.; Fogler, J.; Holzinger, D.; Fellinger, J. Intellectual Disability Profiles, Quality of Life and Maladaptive Behavior in Deaf Adults: An Exploratory Study. Int. J. Environ. Res. Public Health 2022, 19, 9919. https://doi.org/10.3390/ijerph19169919

  1. The study hypothesis needs to be reviewed to tally with the study aim. Please note it is understood that the Independent variables are sensory and physical impairments and the dependent variable is social-emotional functioning. The intellectual disability mainly specifies the participant population

Response: To be clearer the hypothesis now is:

“The hypothesis was that if adults with an intellectual disability have additional sensory impairments such as physical disability or hearing and/or visual impairment they would have lower scores on the SED-S.”

  1. Under participants place the 60.2% just after males to avoid confusing that with the female participants

Response: In the participants section the text is changed to:
This sample included more males (60.2%) than females.

  1. Demonstrate how the SED-S scale is valid and reliable for the adult population as it indicates that it assesses the five developmental stages ranging from 0 to 12 years.

Response: We added the respective information in the method section:

“Expert validity can be taken as given as the scale is based on a survey of develop-mental psychology experts and their assessments of behaviors typical for specific levels of development, [10]. Validation against a group of 160 typically developed children showed a high degree of correspondence (81% exact agreement; 0.95 weighted kappa value; [11]. An exploratory factor analysis provided a one-factor model with a good model fit in 724 adults with ID, most of them having additional mental health problems [15, 41], in 118 children with an ID and mental health problems [9] and in 83 healthy adults with ID [14, 42]. Divergent validity was found for chronological age in children with ID [9]) and in healthy adults with ID [14, 42]. Convergent validity with the Vine-land Adaptive Behavior Scale could be seen in the children's sample (r = .642, p < .001; [9]). Strong negative associations with the severity of ID could be shown in 327 adults with ID and mental health problems (r = -.654, p < .001; [11]), in 83 adults with ID with-out mental health problems (-0.753, p < 0.001; [14, 42]) and in 118 children with ID (G = -.69; p < .001; [9]). Inter-rater reliability for 25 typically developed children was 1.0 (Co-hen's kappa). Internal consistency as measured by Cronbach's alpha was .99 in typically developing children [11], .94 in 118 children with ID [9] and .92 in 83 adults with ID without mental health problems [14, 42].”

           8. Under limitations, it is suggested that instead of saying file studies, rather than write participants' records.

Response: Thank you, indeed. These changes are made.

Author Response

Dear reviewer

We appreciate the feedback we received. We are also grateful for the opportunity to revise our manuscript. Below we will respond point-by -point on each reviewer’s comments. In the revised manuscript, revisions are marked in yellow. The language editing was done this is not marked.

Please the the attachment
